# Radiogenomics in Renal Cancer Management—Current Evidence and Future Prospects

**DOI:** 10.3390/ijms24054615

**Published:** 2023-02-27

**Authors:** Matteo Ferro, Gennaro Musi, Michele Marchioni, Martina Maggi, Alessandro Veccia, Francesco Del Giudice, Biagio Barone, Felice Crocetto, Francesco Lasorsa, Alessandro Antonelli, Luigi Schips, Riccardo Autorino, Gian Maria Busetto, Daniela Terracciano, Giuseppe Lucarelli, Octavian Sabin Tataru

**Affiliations:** 1Department of Urology, European Institute of Oncology (IEO) IRCCS, 20141 Milan, Italy; 2Department of Oncology and Hemato-Oncology, University of Milan, 20141 Milan, Italy; 3Department of Medical, Oral and Biotechnological Sciences, G. d’Annunzio, University of Chieti, 66100 Chieti, Italy; 4Urology Unit, SS. Annunziata Hospital, 66100 Chieti, Italy; 5Department of Urology, ASL Abruzzo 2, 66100 Chieti, Italy; 6Department of Maternal Infant and Urologic Sciences, Policlinico Umberto I Hospital, University of Rome, 00161 Rome, Italy; 7Department of Urology, Azienda Ospedaliera Universitaria Integrata of Verona, University of Verona, 37126 Verona, Italy; 8Department of Neurosciences and Reproductive Sciences and Odontostomatology, University of Naples Federico II, 80131 Naples, Italy; 9Urology, Andrology and Kidney Transplantation Unit, Department of Precision and Regenerative Medicine and Ionian Area, University of Bari Aldo Moro, 70124 Bari, Italy; 10Department of Urology, Rush University, Chicago, IL 60612, USA; 11Department of Urology and Renal Transplantation, University of Foggia, 71122 Foggia, Italy; 12Department of Translational Medical Sciences, University of Naples Federico II, 80131 Naples, Italy; 13Department of Simulation Applied in Medicine, The Institution Organizing University Doctoral Studies (I.O.S.U.D.), George Emil Palade University of Medicine, Pharmacy, Sciences, and Technology of Târgu Mureș, 540142 Târgu Mureș, Romania

**Keywords:** renal cancer, radiomics, radiogenomics, genomics, artificial intelligence, machine learning

## Abstract

Renal cancer management is challenging from diagnosis to treatment and follow-up. In cases of small renal masses and cystic lesions the differential diagnosis of benign or malignant tissues has potential pitfalls when imaging or even renal biopsy is applied. The recent artificial intelligence, imaging techniques, and genomics advancements have the ability to help clinicians set the stratification risk, treatment selection, follow-up strategy, and prognosis of the disease. The combination of radiomics features and genomics data has achieved good results but is currently limited by the retrospective design and the small number of patients included in clinical trials. The road ahead for radiogenomics is open to new, well-designed prospective studies, with large cohorts of patients required to validate previously obtained results and enter clinical practice.

## 1. Introduction

Renal cell carcinoma (RCC) is one of the most common solid tumors in both male and female patients [1]. Early diagnosis, correct treatment choice based on individual risk stratification, accurate prediction of treatment response, and survival are the cornerstones of RCC treatment.

A correct diagnosis, especially for small renal masses, is fundamental for treatment planning. Several studies have pointed out the risk of over-diagnosis and over-treatment in patients with small renal masses [2]. A correct pre-operative diagnosis might reduce the number of useless treatments and their possible harms. With this aim in mind, renal biopsies have been used. Unfortunately, recent series showed an 80% biopsy core diagnostic rate without differences in the re-biopsy outcomes in terms of the quality of the obtained cores for diagnosis. The non-diagnostic rates at re-biopsy remained in the 20% range [3], with such results limiting the diffusion of renal biopsy in clinical practice.

In the last few years, several tumors and hosts’ characteristics have been investigated to better stratify patients according to individual risk characteristics [4]. Based on these characteristics, several tools have been deployed with the aim of achieving an accurate treatment response prediction, including various biomarkers [5] and scoring systems [6]. However, to date, all these tools have shown only fair accuracy [6,7].

A helping hand might arrive from the use of new technologies that combine and analyze a huge amount of information, allowing an important improvement in all these aspects related to RCC management. Radiogenomics integrates a massive volume of quantitative data derived from imaging with individual genetic characteristics [8]. The use of deep learning (DL) allows the building of prediction models to better stratify patients, guide therapy methods, and evaluate clinical results [8].

The aim of the current review of the literature is to provide physicians with a complete and comprehensive review of the most recent studies on radiomics and radiogenomics use in the field of RCC, to emphasize the current knowledge of radiomics, genomics and molecular tumor characterization and its derived molecular imaging, and to identify the future course of radiogenomics-related research. 

## 2. Results

We have identified studies from the PubMed/Medline database, and the workflow of radiogenomics in renal cancer is depicted in Figure 1. The aim was to access original research on the subjects of radiomics, genomics, molecular imaging, and radiogenomics related to renal cancer. Keywords used to search the database were renal cancer, radiomics, radiogenomics, genomics, artificial intelligence, and machine learning (ML). We have included articles up to December 2022 with no time frame limit and, we have excluded non-English, reviews, and case report studies.

### 2.1. Radiomics

Radiomics, which focuses on improving the analysis of large data sets using semi-automatic or automatic software, is a quantitative image analysis of textures and features provided by imaging tools (e.g., multiparametric magnetic resonance imaging (mpMRI)) [9,10]. Numerous malignancies were examined using this model [11,12,13,14,15]. 

Human readers conduct qualitative analysis on radiological pictures. Instead, radiomics seeks to quantitatively map out pictures. Focused on various imaging techniques that are utilized as a starting point, this method is based on the extraction, analysis, and modeling of multiple image elements in connection to specified goals that may be both anatomical and functional [16]. The phases that make up a radiomics investigation include data selection, medical imaging, feature extraction, exploratory analysis, and modeling. The sixteen separate components that make up the radiomics quality score examine every facet of the radiomics process through the five stages mentioned above. The radiomics quality score specifically took into account trial registration, image availability, cut-off and accuracy analyses (calibration and accuracy statistics), multiple segmentations, the phantom study, imaging time points, adjustment for multiple testing, the use of multivariable analyses, the detection and discussion of biological correlates, as well as cost-effectiveness analysis and comparison with the current gold standard [16]. All the processes on which radiomics is based on are called the radiomics pipeline. 

### 2.2. Radiomics in Renal Cancer Management

Radiomics has been developed to assist physicians in improving the diagnosis and management of several oncological diseases. Quantitative evaluation of data from imaging has demonstrated improvements in the diagnostic, prognostic, and predictive roles of conventional radiological images [17,18,19,20]. Current applications of radiomics in RCC management include differentiation of benign from cancerous kidney tumors, differentiation of angiomyolipoma (AML) from RCC, differentiation of oncocytoma (ONC) from RCC, differentiation of different subtypes of RCC, nuclear grade prediction, and evaluation of treatment response of renal masses [21,22,23,24,25,26,27,28,29,30,31,32,33,34,35,36,37,38,39,40,41,42,43,44,45,46,47,48,49,50,51,52,53,54,55,56,57,58,59,60,61,62,63,64,65,66,67,68,69,70,71,72,73,74,75,76,77]. Appendix A summarizes radiomics studies on RCC management according to these specific purposes.

#### 2.2.1. Differentiation of Benign from Cancerous Kidney Tumors, of Angiomyolipoma from RCC, and of Oncocytoma from RCC

Radiomics allows a better characterization of renal lesions in the pre-operative setting and could potentially lead to the avoidance of many unnecessary surgeries for benign lesions. 

ML and DL algorithms have been extensively used in research studies to extract and analyze numerous quantitative features (i.e., histograms, textures, and shapes) from different image modalities. Histogram-based features of skewness and kurtosis from computed tomography (CT) images have been used for the discrimination of benign lesions from cancerous lesions [22]. CT and ML texture analysis based on random forest (RF) algorithm radiomics was used to discriminate RCC from other benign renal lesions [24]. In a larger cohort of renal lesions, CT features were selected and used to build a radiomics ML classification model of RCC versus other renal masses [26]. The best ML algorithm, based on images obtained from 18 different CT scanners, was the RF, which performed better than radiologists’ assessments, highlighting the role of radiomics in limiting the inter-observer and inter-machine variability of standard methods [25]. Similar results have been obtained by an ensemble DL model [30]. Using magnetic resonance imaging (MRI) to gather quantitative and qualitative data and analyze it through artificial intelligence (AI) algorithms had confusing results in different studies [29,30,31,32]. 

#### 2.2.2. Differentiation of Different Subtypes of RCC

The application of radiomics for a better differentiation of clear cell (cc)RCC from other tumor types would allow for a tailored management of RCC. For this specific aim, both CT and MRI features were analyzed in multiple studies. CT images and features, in combination with ML algorithms, were used for differentiating non-ccRCC from ccRCC. The best performance was achieved by an artificial neural network (ANN) classifier [54] and DL neural networks based on CT images along with a ML algorithm, which showed promising results [55] to discriminate ccRCC from papillary (pap)RCC [57].

#### 2.2.3. Nuclear Grade Prediction

The possibility to preoperatively assess the tumor nuclear grade by imaging has been evaluated using numerous studies by both CT- and MRI-based radiomics approaches, which have shown the ability to accurately predict the presence of high- versus low-grade RCC, often by the use of texture analysis [61,62,63,64,65,66,67,68,69,70]. 

#### 2.2.4. Evaluation of Treatment Response 

Diffusion-weighted imaging (DWI)-MRI was evaluated as a possible biomarker for overall survival (OS) in patients treated with sunitinib [72], but OS had no correlation with MRI features. Similarly, integrated positron emission tomography/MRI (PET/MRI) radiomics analysis was used to evaluate the response to sunitinib [71]. Moreover, texture parameters on CT images were assessed as predictive radiomics markers of response to therapy in metastatic (m)RCC patients [73,74,75,76,77], with texture uniformity discovered to be an independent predictor of time to progression (*p* = 0.005) [74], entropy modifications being a good predictor for OS (*p* = 0.02 and *p =* 0.04) and normalized standard deviation (nSD) that can predict progression free survival (PFS) (*p =* 0.01 and *p =* 0.003). The current results cannot definitively lead to the conclusion that radiomics could improve the prediction of treatment response rates.

### 2.3. Genomic and Molecular Tumor Characterization 

Renal cell carcinoma occurs either in sporadic or inherited forms. RCC subtypes are classified according to histopathological features and molecular drivers. Generally, hereditary RCC syndrome is transmitted in an autosomal dominant manner. In cases of de novo germline mutation or incomplete penetrance, a family history of RCC will be lacking. An increased risk of RCC is associated with germline variants of at least 12 genes. Genetic counseling should be offered to patients with multi-centric/bilateral tumors, an early age of diagnosis (<45 years of age), or a first/second degree relative with any kidney cancer. Understanding the genomic features of RCC may lead to a better screening and management of at-risk individuals and improve patients’ prognosis [78,79,80,81,82].

#### 2.3.1. Genetics of Clear Cell-Renal Cell Carcinoma (ccRCC)

The first step in ccRCC development is the loss of the short arm of chromosome 3 (3p loss), frequently because of chromothripsis: multiple breaks will occur in chromosome 3p, followed by a random joining of segments [83]. Four tumor suppressor genes map in this region: VHL in 3p.25, PBRM1 (subunit of the PBAF SWI/SNF chromatin remodeling complex), BAP1 (histone deubiquitinase), and SETD2 (histone methyl transferase) in 3p.21. The latter genes are involved in the maintenance of chromatin status [84,85]. This event will lead to several genomic rearrangements (first hit in tumorigenesis). Later, a “second hit” will provoke the biallelic inactivation of these tumor suppressor genes. Clear cell RCCs are characterized by marked intratumor heterogeneity (ITH), which is referred to as the selection of tumor cell subpopulations with different driver mutations [86]. Remarkably, somatic mutations of VHL are observed in approximately 92% of patients diagnosed with ccRCC, whereas they are not found in non-clear cell RCC. Less frequently, alterations in TP53, mTOR, TSC1, TSC2, PIK3CA, KDM5C, and SMARCA4 are observed in ccRCC [87].

#### 2.3.2. Von Hippel Lindau Disease

Von Hippel Lindau (VHL) disease is a systemic disorder transmitted in an autosomal dominant manner [88,89]. Incidence is estimated 1:34,000 and will likely complete penetrance by age 60. The mean age of onset of ccRCC is 44 years (two decades earlier than sporadic tumors). The risk of metastases is virtually nil when tumors are below 3 cm in size. Besides ccRCC, affected patients develop hundreds of renal cysts, benign pancreatic cysts, central nervous system and retinal hemangioblastomas, and neuroendocrine tumors (NET) such as pheochromocytoma. According to the predisposition to pheochromocytoma, VHL subclasses have been classified. The VHL gene encodes for pVHL, which is part of the E3 ubiquitin ligase complex (VCB complex) with cullin 2 (CUL2), Elongin B and C [90]. Broad spectrums of germline mutations in VHL gene have been described. The VCB complex targets the hypoxia-inducible factors (HIF-1α and HIF-2α). Under normal oxygen conditions, HIF-α becomes hydroxylated on specific sites by HIF-prolyl hydroxylases (PHDs). Molecular oxygen, 2-oxoglutarate (2-OG), and iron are demanded as cofactors. Then, pVHL binds hydroxylated HIF-α through its β-domain, thus targeting its ubiquitination and its proteasomal degradation. Conversely, in the case of hypoxia, the accumulation of HIF-α will lead to the activation of the hypoxia-induced response element (HRE) genes. The loss of VHL leads to the constitutive activation of HRE in the absence of hypoxia (“pseudohypoxia” is typical of both sporadic and hereditary ccRCC). HRE genes enhance altered anaerobic metabolism (GLUT1, PDK1, and EPO), angiogenesis, proliferation, and cell survival (VEGF, PDGF, TGF-α, and cyclin activation). Typically, mutations occur in the pVHL α binding domain to Elongin C [91,92]. 

#### 2.3.3. Genetics of Non-Clear Cell-Renal Cell Carcinoma 

##### Papillary Renal Cell Carcinoma

Papillary renal cell carcinomas (papRCC) have been classically divided into two different subtypes according to their histology and genetics. Sporadic and hereditary forms exist. Type 1 papRCC is characterized by the gain of chromosomes 7 and 17 and the loss of chromosomes 2, 3, 12, 16, and 20. Activating mutations of the MET gene on chromosome 7 are usually observed [93,94,95]. In turn, type 2 papRCCs do not present a specific pattern of copy number alterations. According to The Cancer Genome Atlas Research Network, type 2 papRCC may show CDKN2A silencing, SEDT2 mutations, TFE3/TFEB gene fusion, and increased expression of the NRF2-antioxidant response element pathway. Currently, a heterogeneous group of RCCs with papillary features and more aggressive behavior have been described, including translocation RCCs, FH-deficient and SDH-deficient RCCs. New potential RCC entities include papillary renal neoplasm with reversed polarity (PRNRP) and biphasic hyalinizing psammomatous RCC (BHP RCC), which have distinct driver mutations (KRAS and NF2, respectively) [96,97,98,99].

##### Hereditary Papillary Renal Carcinoma (HpapRCC)

HpapRCC is an autosomal dominant disorder with missense mutations of the MET protoncogene on chromosome 7q. MET encodes the hepatocyte growth factor (HGF) receptor. As a result of these mutations, constitutive activation of HGF’s downstream pathway occurs. Affected individuals are at risk of developing multifocal, bilateral type 1 papRCC. Complete penetrance is approximately described by the age of 80, even if the age of onset is associated with a specific missense mutation. No extrarenal manifestations are observed [100,101,102,103].

##### Chromophobe Renal Cell Carcinoma (chRCC)

A distinct pattern of chromosomal alteration characterizes chRCC. Chromosomes 1, 2, 6, 10, 13, and 17 are lost in approximately 80% of cases of chRCC. Less frequently observed loss involves chromosomes 3, 5, 8, 9, 11, 18, and 21q. The next most common mutations affect TP53, PTEN, CDKN2A (loss of 9p21 or hypermethylation), and TERT. Moreover, increased mitochondrial deoxyribonucleic acid (DNA) copy numbers and increased expression of the mitochondrial regulator PPARGC1A suggest increased biogenesis in chRCC [104,105,106,107].

##### Birt-Hogg-Dubé (BHD) Syndrome 

Birt-Hogg-Dubé (BHD) syndrome is an autosomal dominant cancer predisposition due to germline mutations in the FLCN gene (17p11) encoding folliculin [108,109,110]. Apart from bilateral and multifocal kidney tumors, affected patients develop benign cutaneous fibrofolliculomas and pulmonary cysts (risk factor for spontaneous pneumothorax). No associations between genotype and phenotype exist. Renal tumors display different histological features: hybrid oncocytic tumors (50%), chRCCs (34%), and oncocytomas (9%) are common. Folliculin (FLCN) may be involved in the PI3K/AKT/mTOR pathway through FLCN-interacting proteins FNIP1 and FNIP2 [111,112,113].

##### Renal Medullary Carcinoma

Renal medullary carcinoma (RMC) accounts for less than 1% of kidney cancers. Early metastases are responsible for poor OS. It afflicts predominantly patients with sickle cells trait. Common genetic mutations are the loss of expression of the SMARCB1 protein (SWI/SNF-related matrix-associated actin-dependent regulator of chromatin subfamily B member 1), BRG1-associated factor 47 (BAF47), or sucrose non-fermenting 5 (SNF5). SMARCB1 is part of the SWI/SNF chromatin remodeling complex: its loss deregulates many transcriptional pathways [114].

#### 2.3.4. Additional Types of Renal Cell Carcinoma

##### FH-Deficient and SDH-Deficient Renal Cell Carcinoma

Hereditary leiomyomatosis and renal cell carcinoma (HLRCC) syndrome is inherited in an autosomal dominant manner. Mutations may occur throughout the entire fumarate hydratase (FH) gene on chromosome 1q43. Missense, frame shift, nonsense, splice-site mutations, or complete gene deletions have been described. No genotype-phenotype linkage is observed. Patients are at risk to develop benign skin and uterine leiomyomas and an aggressive form of RCC (formally a type 2 papillary RCC). RCC lesions are typically solitary with rapid tumor growth and early metastatic seeding [115,116,117,118,119,120]. Germline mutations of succinate dehydrogenase (SDH) subunits are associated with hereditary adrenal or extra-adrenal pheochromocytoma (PCT) and head and neck paraganglioma (PGL) syndrome, gastrointestinal stromal tumors (GIST), and RCCs with different histological features. Clinical manifestations depend on the mutated SDH subunit; missense, frame shift, and nonsense mutations are included. Solitary and unilateral RCC are usually diagnosed [121,122,123]. SDH and FH (enzyme of Krebs cycle) loss leads to the accumulation of succinate and fumarate [124]. These oncometabolites competitively inhibit the HIF-PHD interaction, which requires 2-oxoglutarate. This pseudohypoxic phenotype results in HIFα- stabilization and upregulation of HIF-inducible genes [125,126,127,128]. In addition, they block a group of α-KG dependent dioxygenases leading to epigenetic dysregulation. KDM4A and KDM4B dioxygenases blockade causes the suppression of the homologous recombination DNA-repair pathway. Furthermore, the aberrant succination of KEAP1 leads to the upregulation of the NRF2-mediated antioxidant signaling pathway. Moreover, a significant decrease in mtDNA content and an increase in mtDNA mutations have been described due to the accumulation of these oncometabolites [129,130,131,132,133].

##### Translocation Renal Cell Carcinoma (T-RCC)

Translocation renal cell carcinomas (T-RCCs) are driven by somatic translocation involving a member of microphtalmia (MiT) transcription factor family genes on chromosome X (i.e., TFE3, TFEB or MITF) with other partner genes [134]. This gene fusion alters many biological processes such as organelle biogenesis and cell proliferation [135,136,137,138]. They account for the most common forms of RCC in children and young adults. T-RCCs typically present with papillary architecture, even if papillary or clear cell subtypes of T-RCCs have been described in adult patients. 

#### 2.3.5. Renal Cell Carcinoma as a Metabolic Disease

The recent re-evaluation of cancer as a metabolic disorder has led to the discovery of specific oncometabolites with an important role in different processes of tumor biology such as proliferation, progression, and metastatisation [139,140,141,142,143]. In this scenario, recent studies showed that an altered metabolism has a fundamental role in the development of RCC [144,145]. In fact, it has been shown that many genes that are mutated or aberrantly expressed in this tumor also control different cell metabolic activities [144,145].

In particular, in ccRCC, we observe a metabolic reprogramming characterized by an anaerobic switch that induces the rerouting of sugar metabolism toward the pentose phosphate pathway (with the aim of promoting both nucleotide biosynthesis and redox homeostasis) and impairs the mitochondrial activity through the overexpression of NADH dehydrogenase (ubiquinone) 1 alpha subcomplex 4-like 2 (NDUFA4L2) [146,147,148]. NDUFA4L2 is one of the most expressed genes in ccRCC and encodes a protein that reduces mitochondrial oxygen consumption through inhibiting the electron transport chain Complex I. This protein is also involved in additional processes such as cell proliferation, cancer cell migration, and angiogenesis [148]. 

RCC is also characterized by significant accumulations of polyunsaturated fatty acids in association with increased expression of stearoyl-CoA desaturase (SCD1) and FA elongase 2 and 5 [149], as well as metabolic heterogeneity that may allow for subtyping of cancers and prediction of clinical outcomes [150].

### 2.4. Molecular Imaging in Renal Cancer

Molecular imaging (MI) is the branch of radiology that enables the in vivo visualization and quantification of both physiological and disease-specific biological events at the cellular and molecular levels in living organisms. It employs MI probes or contrast agents that specifically interact with molecules to target tissues of interest. In the field of oncology, by allowing the visualization of metabolic processes and specific targets of tumor, differently from conventional imaging methods which only provide structural information, MI has been recently used as non-invasive tool of precision medicine for an optimized cancer management [151]. 

Specifically, for RCC, by targeting imaging using biomarkers involved in processes related to a specific RCC subtype, such as overexpressed proteins or dysregulated biological pathways, MI could not only allow for RCC detection (i.e., distinguishing benign from malignant renal lesions) but also for tumor subtyping; moreover, it could be valuable for staging, prognosis prediction, and treatment response assessment. To date, several radiotracers have been used for these aims, and the main MI modalities for RCC management are summarized in Appendix A.

#### 2.4.1. Single Photon Emission Computed Tomography (SPECT)/CT with ^99m^Tc-Sestamibi

^99^mTc-sestamibi is a lipophilic cationic mitochondrial radiotracer that specifically accumulates in cells with high mitochondrial content and low multidrug resistance pump (MDR) expression. Since ONC is characterized by an elevated mitochondrial content with low MDR expression, while ccRCC by a decreased mitochondrial content with high MDR expression, SPECT/CT with ^99m^Tc-sestamibi has been explored for distinguishing between ONC and ccRCC [152,153,154,155,156,157,158]. In the study by Gorin et al., SPECT/CT with ^99m^Tc-sestamibi showed a sensitivity of 87.5% and specificity of 95.2% for the detection of ONC and hybrid oncocytic/chromophobe tumors (HOCT) [155]. Moreover, SPECT/CT with ^99m^Tc-sestamibi was found to properly identify 91.6% and 100% of ONC and HOCT, respectively [157]. These promising results, together with the fact that it is cost-effective [159] and already approved for other diseases (e.g., myocardial and parathyroid), make this MI modality of particular promise for the management of renal lesions, with a potential translation into clinical practice in the near future.

#### 2.4.2. Positron Emission Tomography (PET)/CT with Radiolabeled Girentuximab

Girentuximab is a humanized monoclonal antibody (mAb) that selectively binds to carbonic anhydrase IX (CAIX), a transmembrane protein of the carbonic anhydrase family that, in physiological conditions, catalyzes the hydration of carbon dioxide in response to hypoxia. CAIX is overexpressed in almost all ccRCCs (>95%), due to inactivating mutations of the VHL gene [160]. Since CAIX can be labeled with numerous radionuclides, radiolabeled girentuximab has been evaluated for ccRCC management. 

In a pilot study by Divgi et al., ^124^I-girentuximab PET/CT correctly diagnosed 93.5% of ccRCC [161]. Subsequently, the same group evaluated its performance in comparison with standard CT imaging for ccRCC diagnosis, and, according to their results, ^124^I-girentuximab PET/CT performed better than conventional imaging (86.2% sensitivity and 85.9% specificity versus 75.5% sensitivity and 46.8% specificity, respectively) for the detection of ccRCC [162]. 

Since I-labeled tracers tend to accumulate in thyroid tissue, other radiotracers have been explored [163]. Several studies have suggested that labeling girentuximab with ^89^Zr can lead to higher tumor uptake and retention and perhaps be more sensitive in the detection of ccRCC than ^124^I. A multicenter prospective trial is ongoing to evaluate the use of ^89^Zr-girentuximab PET/CT for this purpose (ClinicalTrials.gov Identifier: NCT03849118). Moreover, ^89^Zr-girentuximab PET/CT performed better than both conventional CT and ^18^F-FDG PET/TC for ccRCC metastases detection (detection rates of 70%, 56%, and 59%, respectively) [164]. MI targeting CAIX has also been used to assess adjuvant treatment responses. In the study by Muselaers et al., a significant reduction in tumor uptake of ^111^In-girentuximab (38.4% reduction rate) was seen in patients treated with sorafenib [165]. ^111^In-girentuximab was also evaluated with SPECT/TC for the diagnosis of ccRCC, showing a PPV of 94% [166]. However, despite this encouraging result, SPECT is less advantageous than PET as MI modality [167]. Of note, although results with mAb are promising, several barriers to a widespread clinical application of mAb as probes for MI exist, such as logistical issues related to the long timeframe between their injection and image acquisition; for instance, this interval is 3–7 days for girentuximab, while it is only 75 min with ^99^mTc-sestamibi. Thus, although ^124^I- and ^89^Zr-girentuximab have proven value in distinguishing benign from malignant renal lesions, as well as in detecting and monitoring primary and metastatic sites of ccRCC, future studies should focus on new MI agents (e.g., low-molecular-weight agents) targeting CAIX to overcome current limitations. 

#### 2.4.3. PET/CT with ^18^F-FDG

2-deoxy-2-[^18^F] fluoro-D-glucose (^18^F-FDG) is a small molecule radiotracer that targets tumor cells with increased aerobic glycolysis. It is the most used PET radiotracer in oncology and has been largely evaluated for RCC management at different steps, yet it has limited utility in the initial diagnostic workup of renal lesions [168]. Indeed, it is excreted through the kidneys—making the distinction of tumors from the normal renal parenchyma difficult—, and some RCC exhibit a low ^18^F-FDG uptake due to tumor heterogeneity; furthermore, ^18^F-FDG is non-specifically taken up by any malignant cells with cytosolic aerobic glycolysis—thus it is not suitable for proper RCC subtyping [169,170]. The pooled sensitivity for renal mass diagnosis with ^18^F-FDG PET was 62% [171], and the performance did not improve with PET/CT [169,171,172]. 

^18^F-FDG PET/CT has also been evaluated for the diagnosis of recurrent RCC. In the study by Alongi et al., ^18^F-FDG PET/CT showed a sensitivity and specificity of 74% and 80%, respectively (sensitivity and specificity were 88.8% and 70.2%, respectively, with conventional CT) [173]. The pooled sensitivity and specificity for recurrence detection with ^18^F-FDG PET/CT were 92.3% and 97%, comparable to conventional imaging [174]. For distant metastases detection, ^18^F-FDG PET/CT showed a sensitivity and specificity of 92.5% and 99.6%, respectively (compared to 93.3% and 94% with conventional imaging), performing better than CT with regards to lymph node, bone, and soft tissue metastases, whereas CT was superior for lung evaluation [175]. A high ^18^F-FDG PET uptake has also been associated with RCC aggressiveness and worse prognosis (e.g., nuclear grade and sarcomatoid features) [176,177,178]. Finally, ^18^F-FDG PET/CT might play a role in the prediction of treatment response in metastatic RCC patients treated with either TKI or ICI, since metabolic changes related to therapies occur well before structural changes [179,180,181]. In the study by Tabei et al., elevated SUVmax after ICI was an independent predictor of the response to treatment [181]. Of note, ^18^F-FDG PET/CT does not require contrast agents—which is particularly useful for monitoring patients with renal dysfunction—and has a lower radiation dose than CT.

#### 2.4.4. Prostate-Specific Membrane Antigen (PSMA)-Targeted PET/CT

Prostate-specific membrane antigen (PSMA) is a cell surface protein with folate hydrolase activity, which is overexpressed on most prostate adenocarcinomas, yet also in tumor-associated neovascular endothelial cells of numerous solid tumors [182]. With specific regards to RCC, a positive PSMA staining was found in 76.2% of ccRCC neovasculature, and 31.2% of chRCC, whereas papRCC was PSMA negative. Although a few PSMA ligands have been studied for RCC, ^68^Ga-PSMA-11 is the most widely studied PSMA-targeted PET tracer. Due to its high accumulation in the kidney, it has a limited role in the characterization of renal lesions. However, after a case report demonstrated ^68^Ga-PSMA-11 PET/CT could detect RCC metastases, several studies have demonstrated its superiority in the detection of RCC metastases (sensitivity of 92.1% and PPV of 97.2%) when compared to CT (sensitivity of 68.6% and PPV of 80%) [183]. This MI modality was also explored for RCC aggressiveness assessment. SUVmax was associated with high nuclear grade and adverse pathology (e.g., tumor necrosis and sarcomatoid features) [184]. To date, MI using PSMA still requires further research and comparative clinical trials; furthermore, the development of new compounds, such as tracers with hepatobiliary clearance or with improved renal clearance, may optimize the performance of the RCC workup. 

#### 2.4.5. PET/CT with ^11^C-Acetate

Acetate is an important substrate for energy metabolism in the tumor since it is involved in numerous metabolic processes, including lipid synthesis, which is significantly upregulated in RCC. However, RCCs show a high heterogeneity for acetate uptake, and the data are still conflicting in the literature. For instance, in the study by Oyama et al., ^11^C-acetate PET was found to differentiate between malignant lesions—which absorbed more acetate—and benign masses [185], whereas in the study by Kotzerke et al., RCC did not concentrate acetate [186]. Interestingly, in the study by Ho et al., RCC patients underwent ^11^C-acetate and ^18^F-FDG dual-tracer PET/CT, and a higher acetate uptake was seen in benign renal masses than in malignant [187]. With regards to specific patterns of uptake by different RCC subtypes, chRCC was found to take up only acetate, papRCC only FDG, while ccRCC FDG and acetate for high- and low-grade tumors, respectively. Dual-tracer PET/CT may help with RCC subtyping. The need for two PET studies due to the short half-life of ^11^C and the limited evidence that we have in the literature make this MI modality far from application into clinical practice. Thus, the future application of such technologies will widely depend on the current paradigm of screening and diagnosis [188], which still will play a key role in the decision making for surgical scenarios [189] and in the prediction of perioperative outcomes in RCC management with the possible and promising integration of haemato-chemical biomarkers, which might improve molecular imaging accuracy as it has been previously demonstrated in the diagnosis and monitoring of other several genito-urinary (GU) diseases and malignancies [20,190,191,192,193,194]. 

### 2.5. Radiogenomics in Renal Cancer Management 

The approach of combining radiomics features and gene expression in renal cancer has achieved new heights in the last few years. Almost all of the studies had pursued this approach in clear cell renal cell carcinoma. This is possible in some way due to the small number of mutated genes in ccRCC, which can translate into a direct association of radiomics features and genomics [195]. The proper management of ccRCC is closely related to its diagnosis, treatment, follow-up, and prognosis. Diagnosis using imaging techniques and its advancements in characterizing the obtained cross-sectional images is still struggling to better predict the ccRCC subtypes and their underlying molecular and gene characteristics and has its limitations [196,197]. The rise of AI and the related machine and deep learning models and algorithms, combined with an increase in computational power and adjusted statistical models that are able to process the high amount of data coming from radiology that are extracted by conventional imaging, were combined with genomic data and therefore gave birth to a new and exciting new area of research, radiogenomics [198,199]. Treatment is also posing challenges when it comes to establishing the prognosis and prediction of evolution, especially in the early stages of the disease, because of the genetic modifications that could potentially alter the personalized and individualized choice of treatment. It is well known that the intra-tumoral molecular heterogeneity of ccRCC can lead to an incomplete diagnosis, poor choice of treatment, and errors in predicting the post-therapy evolution of the disease [195,200,201,202]. By assessing the whole tumor molecular pattern, radiogenomics can improve the tumor characterization because it analyzes a broader tissue sample. The combined imaging phenotypes and genetic data (genotypes) can boost biomarkers discovery that can predict tumor response to a certain therapy or enrolling the patient in an active surveillance program [203]. The current published research that aimed to analyze if radiomics features are associated with mutational status, could predict mutated genes or the one that goes beyond molecular characterization, that of establishing clinical outcomes through radiogenomics, are summarized in Appendix A.

Radiogenomics is a novel field of research in many types of cancer. Researchers are just at the beginning of discovering the relation between radiomics features and genomics-related gene signatures to better characterize tumors. In RCC and mostly in ccRCC, the studies performed so far have aimed to identify the link between radiomics features, or phenotypes, and genomics features, or genotypes [195]. 

The research began with the aim of identifying if radiomics features are associated with mutational status in RCC. As early as 2014, Karlo et al. [204] in a retrospective analysis aimed to identify the relationship between phenotypes and gene mutation status and found that VHL, KDM5C and BAP1 SETD2, KDM5C and BAP1 mutations correlates with radiomics features and that the last three mentioned mutations were absent in multicystic lesions, and VHL and PBRM were associated with solid lesions [204]. Following studies also aimed to identify a relationship of radiomics features with genetic expression for BAP1 and PBRM1 [205,206,207,208,209], molecular subtypes [210], whole-transcriptome sequencing (WTS) [211], CT radiomics subtypes [212], gene modules associated with radiomics features [213], or microribonucleic acid (miRNA) expression [204,214,215,216,217]. Some clinical outcomes have been explored, such as OS [212,218,219,220,221,222,223,224], DSS [225] and metastasis prediction [211,213]. Although this research is mandatory in the early phases of a novel area, these studies achieved good and statistically significant results in the correlation between the above-mentioned parameters [204,205,214]. Up to this point, there are a number of advantages and drawbacks that limit the translation of radiogenomics into clinical practice. Advantages are coming from the increased use of AI statistical models (from ML and DL algorithms) [206,226], the new and ongoing discovery of genes that have clinical implications in ccRCC, the innovative research of radiomics in the characterization of extracted features from different scanner platforms (CT, multiphase CT, contrast-enhanced computed tomography (CECT), MRI, PET-CT, and PET-MRI), and the agreement between expert radiologists to create standard protocols and scanners to duplicate the results obtained in pilot studies [195]. Another advantage is that ccRCC has a high landscape of genetic mutations, which can be better assessed as a whole by CT [204], but these mutations are difficult to sample for whole-genome sequencing [204]. Limitations of radiogenomics are due to the existence of many retrospective studies, a limited number of patient data sets, and a few pilot prospective studies due to the imaging methods currently in use, which are performed with different scanners with no standardization of protocols for obtaining the images and little or no assessment of the sensitivity and specificity of radiomics features [203,204]. Multivariate models of mutational status have not been performed in some studies [214] due to the low number of enrolled patients with identified gene mutations. Studies are an incipient discovery phase with little or without external validation of their results, counting as little as just identifying the results of mutations or no mutations (ccRCC exhibiting a high degree of intra-tumoral heterogeneity with altered genomics [227]). Many of the images obtained from scanners are from daily practice and not from well-designed clinical trials, further limiting the reproducibility of results [215]. Associations between imaging features and the mutational status of ccRCC identified that poor edges of the tumor and calcifications are associated with the BAP1 mutation [214] and predict the RUNX3 methylation level [224], and well-defined tumor edges are associated with the VHL mutation [204], as well as the m1 mutation subtype and the less accurate m3 subtype, which is negatively associated with well-defined margins [215]. It is well known that renal vein and urine collector system invasion are predictors for infiltrative phenotypes and are negatively associated with the m3 subtype [215] and positively associated with KDM5C and BAP1 mutations [204]. An exophytic development of ccRCC has been associated with the MUC4 mutation [214] and the VHL mutation with a nodular appearance of ccRCC tumors [204]. CT texture parameters and the association with miRNA expression profiles have been studied by Marigliano et al. [216] and were found to have a weak association using the computer tomography texture analysis (CTTA) samples. Texture analysis is being studied as it has shown good results in predicting OS and response to therapy in different tumor types [228]. Entropy was found to be slightly positive and associated with miR-21-5p expression, and to date there is no explainable reason for this association, except probably the young age of patients and hyper-expression of miRNAs [216]. Vascularity of tumors has been assessed in studies, in order to correlate the tumor enhancement with gene expression, and it was found that brute enhancement of vascularity is associated with VHL mutation [204], high RUNX3 methylation [224], and good PFS in patients receiving tyrosine kinase inhibitors (TKI) and MRI high vascularity [229]. 

The prediction of gene mutation status has been assessed in numerous works since 2015 [205]. Authors highly used ML techniques to compare radiomics features and to predict mutation of different genes (BAP1, PRBM1, or molecular subtypes of ccRCC such as ccA and ccB) [205,206,207,208,209,210]. These studies can be seen as an initial and preliminary effort to discover the potential of radiogenomics and how we can manage renal cancer using this technique. All studies were of retrospective design with a limited number of patients and identified a vast range of AUCs (from 0.52 to 0.987) [205,206], but with relatively good sensitivity, specificity, and accuracy for prediction of gene mutation status. Having these in mind, we can discuss the potential of radiogenomics that can add more value in the management of renal cancer diagnosis and follow-up, but there is still a long way to go before implementation in clinical practice.

### 2.6. Current Challenges, Limitations and Future Perspectives

Radiogenomics has elevated the interest of many disciplines in the medical sciences due to the possibility to correlate imaging with genomic data, aiming to reach the concept of tailored and personalized medicine [230,231]. Nevertheless, considering the mechanisms of gene expression and signaling pathways, the relationship between imaging features and genomic data could be biased, and no proper application in the clinical setting has been established [232]. Given the complex process represented by the radiomics flow, one of the main limitations of radiogenomics is that the large amount of data obtained by extracted radiomics features may cause over-fitting in the radiogenomics model when compared to diagnosed genetic mutations [233]. The discrepancies between the dimensionality of imaging and the whole-genome sequencing or molecular profile are quite evident; indeed, it has to be stated that only a handful of studies could be considered real radiogenomics studies in the sense that whole-genome data was used [8]. Another limitation and bias is instead related to the inter-observer variability, which is added to the issue of manual or semiautomatic image segmentation, as well as the reasonable lack of standardization of protocols and scanners among centers [199,234]. The automatic extraction of features is still underpowered compared to manual or semiautomatic segmentation of regions of interest (ROIs), albeit this could also be explained by the limited implementation of automatic imaging feature extraction in the studies [235]. Additionally, the heterogeneity of results and the high cost of genomic testing make the design of prospective studies quite difficult, and a potential temporary solution could be the use of public data resources such as The Cancer Genome Atlas (TGCA) [236]. Another relative limitation is the small sample size of patients involved, which is partly linked to the difficulty in obtaining appropriate imaging and adequate tissue samples for genomic analysis from the same cohort. This limitation also applies to the concept of validation cohorts, which could be underpowered [237].

As a result, the current challenges of radiogenomics are mainly focused on resolving the aforementioned issues, and a helpful aid could be represented by the increasing role of artificial intelligence and deep neural networks, which could allow the combination of genomic, transcriptomic, proteomic, and metabolomic data in a multidimensional manner in order to limit the reported bias [203,231]. The possibility of accessing the large public databases of imaging and genome data could further improve the standardization and management of radiogenomics. In particular, future research in this field should be aimed to increase the size of involved cohorts and provide prospectively built evidence. Future studies should aim to incorporate imaging phenotypes and molecular signatures, constructing clinical trials aimed at further assessing this relationship. In Table 1, we have introduced a summarization of the currently known advantages and limitations of the use of radiogenomics in the management of RCC. 

## 3. Discussion

The role of radiogenomics in renal cancer and in its most common subtype, clear cell renal cancer, is appealing and promising. Considering that the evaluation of a renal lesion is routinely based on CT or MRI images, the possibility to characterize a potentially malignant lesion in terms of genetic, epigenetic, and pathologic heterogeneity via a non-invasive methodology is an undoubted advantage [232]. As reported in our review, the most common somatic gene mutations in renal cancer are related to VHL, PBRM1, BAP1, and SETD2, although a relatively small number of other mutated genes are reported to be involved in kidney cancer. In addition, considering the high intratumoral mutation heterogeneity that has been observed in this malignancy, it is probable that other genetic mutations could be found in a subset of renal malignancies [239]. In this scenario, the role of radiogenomics could be far more valuable than other standard methods of histopathology assessment (such as fine needle aspiration biopsy, for example) [240,241]. Naturally, the method of assessment of gene mutations in radiogenomics is based on different imaging characteristics that are associated with a gene rather than another [199]. VHL mutations, for example, are associated with defined tumor margins, nodular tumor enhancement, and intratumoral vascularization, while a greater vein invasion is significantly associated with BAP1 mutations [56,209]. The successive step is to assign a prognosis inferred from the expression of mutated genes in the analyzed tumors. Another great advantage related to the use of radiogenomic resides in the possibility to limit the inter-observer variability among radiologists, further adding another strong point for radiogenomics utilization in the clinical practice. 

The novelty of radiomics and radiogenomics could provide a more objective interpretation of extracted images. In particular, an important percentage of renal lesions, accounting for up to 30% of diagnosed lesions, are completely benign at histopathology [241]. It is consequential that the use of these models could greatly impact the management of these lesions. In addition, the possibility to obtain gene profiling from imaging could permit not only to identify aggressive lesions from other indolent lesions but also to tailor the treatment for every individual patient. Finally, the possibility to obtain genetic profiles for every scanned lesion could, in addition, improve the clinical research toward the role of genes and epigenetic changes involved in renal carcinogenesis [203]. 

Albeit the majority of studies have focused on developing models for the prediction of mutational profiles, the current panorama is gradually shifting towards the prediction of gene expression patterns as well as epigenetic changes within the tumor and tumor microenvironment. The use of these models may complement the management of localized renal tumors, in particular regarding the clarification of risk profiles of examined tumors. The clinical applicability of radiogenomics remains however limited by several factors such as the limited cohorts of patients involved or the use of the same publicly available cohort, as the TGCA, which could overfit the predictive models. Secondly, considering that radiomics and radiogenomics studies are heavily dependent on the quality of acquired images, the difference in technical factors such as CT scanners, acquisition modes, and voxel reconstruction algorithms could represent another limitation of these models. Finally, the extraction of radiomics features is still too heterogeneous in terms of automated/semi-automated software and ROI delineations [237,242]. 

Radiogenomics, before it can enter clinical practice, must be able to predict the clinical outcomes of patients with RCC, such as OS, DSS, PFS, and metastasis prediction [211,212,213,218,219,220,221,222,223,224,225,229]. Radiogenomics biomarkers and nomograms have been constructed for accurate survival and prognostic prediction of ccRCC, with good AUCs (0.91) for the association of hypoxia-related genes and radiomics features that predict survival and were also externally validated to increase the prognostic power by combining radiogenomics information with clinical and pathological factors [222,223]. The limitations of the current field research are due to the small number of patients and the retrospective nature of certain studies, with only a few that assess outcomes prospectively [220,229]. Most of the data has been obtained through the TCGA and TCIA portals, with little potential for model overfitting and limited external validation [222].

## 4. Conclusions

Radiogenomics research in kidney cancer represents a promising and constantly developing field. The possibility of associating imaging features with gene expression could heavily modify the clinical practice and the surgical approaches to renal lesions. Nevertheless, the road toward this objective is still under construction and further and larger studies are required in order to improve the applicability of radiogenomics and limit the current pitfalls related to small cohorts and heterogeneity of data acquired.

## Figures and Tables

**Figure 1 ijms-24-04615-f001:**
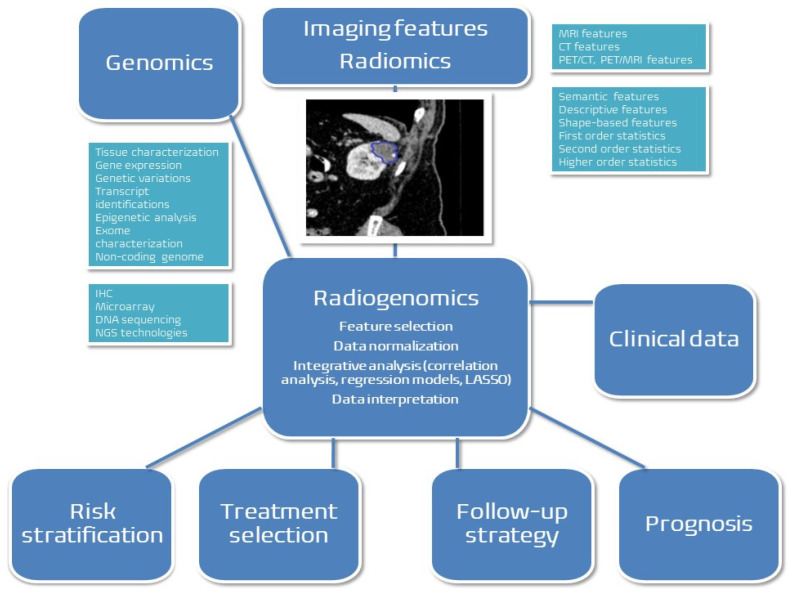
Radiogenomics combining radiomics from imaging features, genomics and clinical data with the purpose of improving clinical decision and patient selection for clinical studies (MRI = magnetic resonance imaging; CT = computed tomography; PET = positron emission tomography; IHC = immunohistochemistry; DNA = deoxyribonucleic acid; LASSO = least absolute shrinkage and selection operator; NGS = next-generation sequencing).

**Table 1 ijms-24-04615-t001:** Advantages and limitations of renal radiogenomics.

Radiogenomics	Advantages	Limitations
	Improvement in screening and management of at-risk individuals [78,79,80,81,82]	Many retrospective studies [203,204]
	Assessment of whole tumor molecular pattern [203]	No standardization of protocols [199,203,204,234]
	Increasing use of artificial intelligence in statistical models [206,226]	Lack of proper clinical trials [215]
	Prediction of overall survival and metastasis [211,212,213,218,219,220,221,222,223,224,225]	Lack of multivariate models of mutation status [214]
	Possibility to tailor treatment according to genomic expressions [230,231]	Inter-observer variability [8]
	Genetic and epigenetic profiling of scanned lesions [203]	Automatic extraction of features still underpowered [235]
	Preoperative identification of benign lesions [238]	High cost of genomic testing to validate data [236]

## Data Availability

No new data were created or analyzed in this study. Data sharing is not applicable to this article.

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
