# Peer review of "Radiogenomics in Renal Cancer Management—Current Evidence and Future Prospects"

_ijms, 2023, doi:10.3390/ijms24054615_

Round 1
Reviewer 1 Report
This review covers the potential of radiogenomics in a new era of personalized medicine, with a strong background about the work that has been previously done that lead to this new research field and all the future possibilities ahead. In fact, radiogenomics has the potential to be used in several cancer models and improve tumor characterization through the combination of the data extracted from both conventional and imaging genomics. The authors did a good job reviewing the work done in this area regarding Renal Cell Carcinoma.
However, some minor issues must be addressed:
1) the authors should review the manuscript and pay attention to english language and grammar. Some of the sentences in the manuscript weren't correct grammatically;
2) Figure 1 quality needs to be improved.
3) Since the manuscript is very extensive, the authors could include a final figure/scheme resuming the potential of radiogenomics in RCC management.
Author Response
We thank the reviewer for the kind words related to our work.
Point-to-point response
1) the authors should review the manuscript and pay attention to english language and grammar. Some of the sentences in the manuscript weren't correct grammatically;
We thank the reviewer for suggesting the above and we had used the services of a professional and native English speaker to improve the retaining of the messages and English language.
2) Figure 1 quality needs to be improved.
We thank the reviewer for this suggestion and we had performed additional graphical improvements for Figure 1.
3) Since the manuscript is very extensive, the authors could include a final figure/scheme resuming the potential of radiogenomics in RCC management.
We thank the reviewer for the suggestion and we hand introduced a Table summarizing the advantages and limitations of radiogenomics in renal lesions management.
Reviewer 2 Report
The review paper "Radiogenomics in Renal Cancer Management – Current Evidence and Future Prospects" by Tataru et al. is well-written, fluid, and concise. The references are well-selected according to the subjects of the review. Several sections highlight the importance of the topics, all with good references and articulation of ideas. Also, the advantages and disadvantages of the techniques are representative of the areas of study of the review. The paper is original, with a high impact in the area of renal cancer and radiogenomics. It clarifies well the reader about the current state-of-the-art techniques involved in renal cancer management. Overall, I have no suggestions to the authors for improvement of the manuscript and I also have not detected major spelling errors or mistakes.
Author Response
We thank the reviewer for the kind words related to our work.